# Demonstration of hypergraph-state quantum information processing

Jieshan Huang [1,7], Xudong Li[1,6,7], Xiaojiong Chen[1], Chonghao Zhai[1], Yun Zheng[1], Yulin Chi[1], Yan Li [1,2,3,4,5], Qiongyi He [1,2,3,4,5], Qihuang Gong [1,2,3,4,5] & Jianwei Wang [1,2,3,4,5] ✉

Complex entangled states are the key resources for measurement-based quantum computations, which is realised by performing a sequence of measurements on initially entangled qubits. Executable quantum algorithms in the graph-state quantum computing model are determined by the entanglement structure and the connectivity of entangled qubits. By generalisation from graph-type entanglement in which only the nearest qubits interact to a new type of hypergraph entanglement in which any subset of qubits can be arbitrarily entangled via hyperedges, hypergraph states represent more general resource states that allow arbitrary quantum computation with Pauli universality. Here we report experimental preparation, certification and processing of complete categories of four-qubit hypergraph states under the principle of local unitary equivalence, on a fully reprogrammable silicon-photonic quantum chip. Genuine multipartite entanglement for hypergraph states is certified by the characterisation of entanglement witness, and the observation of violations of Mermin inequalities without any closure of distance or detection loopholes. A basic measurement-based protocol and an efficient resource state verification by color-encoding stabilizers are implemented with local Pauli measurement to benchmark the building blocks for hypergraph-state quantum computation. Our work prototypes hypergraph entanglement as a general resource for quantum information processing.

Graph states are the multiqubit entangled states, on which universal measurement-based quantum computation (MBQC) can be carried out by applying a sequence of measurements on qubits[1,2]. Figure 1a shows a graph state $|G\rangle = (V, E)$, in which one vertex ($V$) represents one qubit and one edge ($E$) represents one pairwise entangling interaction[2]. Mathematically, hypergraph is a generalization of graph. In quantum physics, a generalization of the graph state is the hypergraph state $|HG\rangle$ in Fig. 1b, in which a hyperedge ($HE$) can arbitrarily entangle a subset of multi-qubits in $V$[3–6], rather than the two nearest qubits in the

graph state in Fig. 1a. The graph state is a special case of the two-uniform hypergraph states. Processing such generalized hypergraph states features interesting nonlocal properties[7–9] and enables unique capabilities in quantum computation[10–13]. For example, hypergraph-state quantum computation achieves Pauli universality, meaning it only requires Pauli measurements for MBQC[11]. It could allow the emulation of non-trivial phases such as symmetry-protected topological order in condensed matter physics[12,13]. The utilization of hypergraph states could also change the circuit-depth complexity and gain

[1]State Key Laboratory for Mesoscopic Physics, School of Physics, Peking University, 100871 Beijing, China. [2]Frontiers Science Center for Nano-optoelectronics and Collaborative Innovation Center of Quantum Matter, Peking University, 100871 Beijing, China. [3]Collaborative Innovation Center of Extreme Optics, Shanxi University, Taiyuan 030006 Shanxi, China. [4]Peking University Yangtze Delta Institute of Optoelectronics, Nantong 226010 Jiangsu, China. [5]Hefei National Laboratory, Hefei 230088, China. [6]Present address: John A. Paulson School of Engineering and Applied Sciences, Harvard University, Cambridge, MA, USA. [7]These authors contributed equally: Jieshan Huang, Xudong Li. ✉e-mail: jww@pku.edu.cn

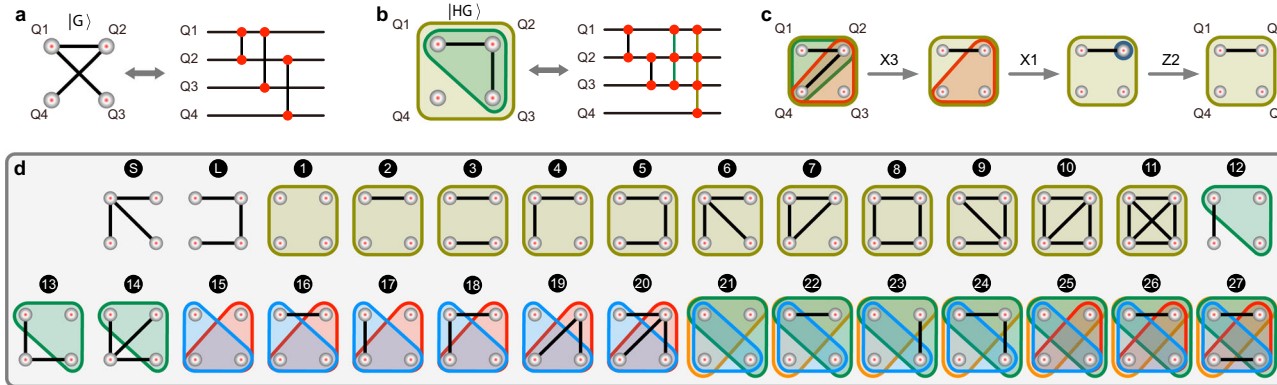

**Fig. 1 | Enumeration of all four-qubit hypergraph states under local unitary transformation. a** Topology of a four-qubit graph state, and **b** a four-qubit hypergraph state. Vertices ($V$) represent qubits (Q1–Q4), and edges ($E$, black lines) represent entangling interactions between two qubits, and hyperedges ($HE$, colored close shapes) represent entangling interactions between multiple qubits. Right plots: Quantum circuits for the preparations of graph state and hypergraph state, shown in the left plots respectively. Vertical colored lines connecting red dots represent the CZ, CCZ and CCCZ entangling gates. **c** An example to show the principle of local unitary (LU) equivalence. The hypergraphs are locally equivalent if they can be mutually transformed by repeatedly applying the local unitary operations on single-qubits, e.g, local Pauli operations X($k$) and Z($k$) on the qubit $k$. The $X(k)$ operation on the qubit $k$ removes or adds these hyperedges in $E^{(k)}$ depending on whether they exist already or not, where $E^{(k)}$ represents all hyperedges that contain qubit $k$ but removing qubit $k$ out. The $Z(k)$ operation on the qubit $k$ removes the one-edge on the qubit $k$. step1 to step 2: the $X3$ operation on Q3; step2 to step 3: the $X1$ operation on Q1; step3 to step4: the one-edge is removed by a $Z2$ operation. **d** Enumeration of all 27 four-qubit hypergraph states that are equivalent under LU transformation. The first two refer to two classes of star ($|S\rangle$) and line ($|L\rangle$) graph states.

gates parallelization[10]. Recently, technological progress has lead to creations of multiqubit graph states in the photonics[14], superconducting[15], trapped ion[16], and atomic systems[17]. By using integrated photonic quantum technologies[18,19], it has enabled the on-chip generation and manipulation of graph states[20–23]. However, the preparation of hypergraph states is rather challenging due to its requirement of implementing a sequence of multiqubit entangling gates.

In this work, we report the preparation, verification and processing of arbitrary four-qubit hypergraph states on a reprogrammable silicon-photonic quantum chip. We realize the complete classes of 27 different four-qubit hypergraph states and 2 different graph states. The Mermin test provides an experimental tool to verify the presence of multiqubit entanglement, by which we verify entanglement without any closure of the distance or detection loopholes. Genuine multiqubit entanglement for hypergraph states is also certified by characterizing entanglement witness. We benchmark the implementation of hypergraph MBQC protocol, providing the building blocks for blind quantum computing with Pauli universality.

## Results

### Theory and scheme

Hypergraph states are $n$-qubit stabilizer states, which correspond to mathematical hypergraphs with vertices ($V$) and ($2^{2^n-1}$) hyperedges ($HE$) in total[3–6]. It can be described as:

$$|HG\rangle = \prod_{e \in HE} C_e |+\rangle^{\otimes n}, \quad (1)$$

where $C_e = I - 2|1...1\rangle\langle 1...1|$, adding the $C^m$-Z ($m < n$) operations on the qubits $|+\rangle = (|0\rangle + |1\rangle)/\sqrt{2}$ connected by the hyperedge $e$. When $C_e$ is chosen as the $CZ$ gate, it returns to the scenario of graph states[1,2]. The preparation of the hypergraph states requires multiqubit entangling interactions that correspond to the hyperedges connecting more than two qubits. Figure 1b shows the quantum circuit for the preparation of a hypergraph state, consisting of a sequence of $C^m$-Z gates. The $C^m$-Z returns to the Toffoli gate for $m = 2$, and to the CZ gate when $m = 1$. The hypergraph states are real equally weighted states, and they can be mapped to the boolean functions with the dual degeneracy of a global

phase factor. For the four-qubit hypergraph states, there are total ($2^{2^4-1}$) enumerations. However, many of them are locally equivalent. Any hypergraph states can be defined as locally equivalent if they can be mutually transformed by repeatedly applying local unitary (LU) transformations[7,9]. The rule of LU transformation is provided in Methods and Fig. 1c caption. As a result, for the four-qubit case, there are 27 different LU-classes of hypergraph states together with 2 LU-classes of graph states. Figure 1d enumerates the 29 classes of hypergraph and graph states. The LU equivalent states share the same degree of entanglement, since only single-qubit operations are performed locally on the qubits.

Using conventional quantum photonic methods, preparing hypergraph states by operating the $C^m$-Z gates (Fig. 1b) is significantly challenging. For example, implementing the three-qubit Toffoli gate requires six CNOT gates. We propose a method to produce the hypergraph states with a high successful probability. In this scheme, quantum entanglement is translated from the photon sources to the entangling gates, by which the difficulty of realizing the $C^m$-Z gate is mapped to the generation of multidimensional (the local dimensionality of $d$ dimensions) multiphoton (the number of $n$ photons) entangled states[24,25]; each qudit is mapped to $\mathrm{Log}_2(d)$ qubits, among which arbitrary entangling gates can be applied determinately. Details and scalability analysis of this method are provided in Supplementary Note 2. In the experiment, we implement the scheme to realize four-qubit hypergraph states on the quantum chip that translates four-dimensional two-photon entanglement to the $C^m$-Z gates, $m \leq 3$.

### A programmable silicon-photonic quantum chip

Figure 2a reports a quantum photonic chip that allows arbitrary on-chip preparation, operation, and measurement of four-qubit hypergraph states. The quantum chip shifts the task of constructing a generic multiqubit entangling gate to operating multipartite entanglement from parametric nonlinear sources[26,27]. We implement a qudit-qubit mapping in which the four-qubit states are encoded into two-ququart states of photons and arbitrary entangling operations can be performed[28]. The device consists of four main parts, i.e, entanglement-generation, space expansion, local unitary operation, and coherent compression, as shown in Fig. 2a. A four-dimensional Bell state is first generated in integrated four-wave-mixing sources[29], which

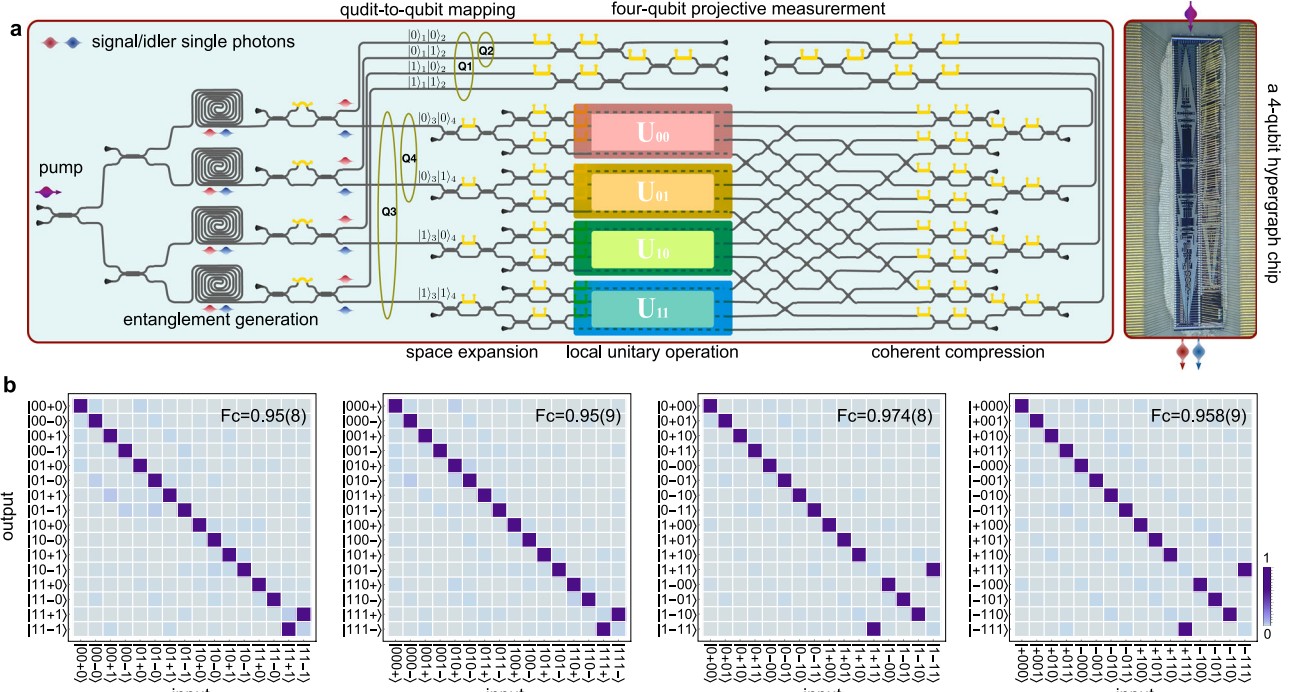

**Fig. 2 | A reprogrammable silicon-photonic quantum chip for the arbitrary preparation, operation, and measurement of four-qubit hypergraph states.** **a** Circuit diagram. The four-qubit hypergraph device is fabricated on a silicon-nanophotonic quantum chip. The chip integrates more than 400 components for the generation, operation, and measurement of four-qubit {Q1, Q2, Q3, Q4} hypergraph states. Two-qubit states are mapped to a four-dimensional qudit state in one single-photon as: $|00\rangle_{qubit} \rightarrow |0\rangle_{qudit}$, $|01\rangle_{qubit} \rightarrow |1\rangle_{qudit}$, $|10\rangle_{qubit} \rightarrow |2\rangle_{qudit}$, $|11\rangle_{qubit} \rightarrow |3\rangle_{qudit}$. The multi-qubit controlled gate $C^m$-$Z$ is enabled by a process of "entanglement generation--space expansion--local operation--coherent compression". Reconfiguring the unitary operations $U_{ij}$ ($i,j = 0,1$) allows the

implementations of $C^m$-$Z$ gates. A sequence of controlled gates are compiled by setting $U_{ij}$ accordingly. Right inset: a photograph for packaged silicon-photonic quantum chip, which is wire bonded (gold lines) on a printed circuit board. **b** Characterization of the CCCZ gate. Truth tables are measured in four different input-output sets of conjugate product bases. {$|0\rangle$,$|1\rangle$} represents the computational basis, and $|\pm\rangle = (|0\rangle \pm |1\rangle)/\sqrt{2}$ represents the Hadamard basis. The probability distributions are coded by colors and the key is provided at the right bottom. The classical statistic fidelity (Fc) shown in the plots is used to characterize each truth table. The CCCZ gate is characterized by the Hofmann fidelity estimated from four tables in **b**.

results in a four-qubit state as $(|0000\rangle + |0101\rangle + |1010\rangle + |1111\rangle)/2$. For the idler single-photon, its operational space is subsequently expanded for applying the arbitrary local unitary operation $U_{ij}$ which is physically realized by universal linear-optic circuits[30]. The entire process retains quantum coherence by space compression. The chip allows implementations of multi-qubit controlled unitary $C^m$-$U$ gates. For example, the triply-controlled CCCZ gate can be obtained by setting the configuration as $U_{00} = U_{01} = U_{10} = II$ and $U_{11} = CZ$. The circuit thus results in the $(|00\rangle\langle00|II + |01\rangle\langle01|II + |10\rangle\langle10|II + |11\rangle\langle11|CZ)$ gate, which functions as the CCCZ gate. Similarly, other $C^m$-$Z$ gates ($m \leq 3$) for generating the hypergraph states (see Fig. 1) can be realized or compiled in the device. Details for state evolution, device fabrication and experimental setup have been provided in Methods.

As an example, we characterize the multiqubit CCCZ entangling gate, by employing an efficient method of generalized Hofmann fidelity[31]. It requires only $4 \times 2^4 = 256$ measurements, consisting of four measurement settings in the partially conjugate product bases. In each setting, three of the qubits are prepared and measured in the Pauli-Z basis $|0\rangle$,$|1\rangle$, while one qubit is prepared and measured in the Pauli-X basis $|\pm\rangle$. Figure 2b shows measured input-output truth tables for the CCCZ gate. The CCCZ gate flips the phase only when the qubits are in the $|1111\rangle$ state. A classical statistic fidelity ($F_c$) is defined to quantify the results, $F_c = \frac{1}{2^4}\sum_{i=1}^{2^4} p_i q_i$, where $p_i, q_i$ are theoretical and measured distributions, respectively. They are measured to be $F_{c1} = 0.95(9)$, $F_{c2} = 0.95(8)$, $F_{c3} = 0.974(8)$, and $F_{c4} = 0.958(9)$, demonstrating high-fidelity CCCZ gate. This gives the lower bound Hofmann fidelity of the gate $F_c > F_{c1} + F_{c2} + F_{c3} + F_{c4} - 3 = 0.84(9)$.

## Preparation and verification of hypergraph states

Executing a sequence of multiqubit $C^m$-$Z$ gates, similar to quantum circuits in Fig. 1b, we create all 29 LU-classes of hypergraph and graph states as shown in Fig. 1d. Note that quantum circuits are compiled in experiment by means of combining the sequence of gates, as the $C^m$-$Z$ operators commute with each other. We reconstruct density matrices for all these states by implementing quantum state tomographical measurements with compressed sensing techniques[32]. Quantum state fidelities $F_q$, that is defined as $F_q = (Tr[\sqrt{\sqrt{\rho_0} \cdot \rho \cdot \sqrt{\rho_0}}])^2$, are estimated from the measured ($\rho$) and ideal ($\rho_0$) density matrices. Figure 3a summarises the estimated fidelities $F_q$ for all LU-classes of hypergraph states and graph states. Though the standard tomographic approach becomes impractical when increasing the number of qubits or at a low photon counting rate, more efficient approaches are available for state verification and fidelity estimation[33].

We measure alternatively the entanglement witness to verify the presence of genuine multiqubit entanglement of the hypergraph states, where all subsystems are genuinely entangled. The witness operator is defined as $\hat{W} = I - \frac{1}{\alpha}|\psi\rangle\langle\psi|$[34,35], where $\alpha$ is the maximum overlap between the given state $|\psi\rangle$ and product states in any bipartition. Measuring the expectation value of witness provides an efficient way for entanglement verification. As an example, we verified the hypergraph state no.11 in Fig. 1d. We decompose the state into a linear combination of products of single Pauli operators and measure their average values. Experimental results are reported in Fig. 4, in good agreement with theoretical ones. The measured witness value is $\langle\hat{W}\rangle = -0.42 \pm 0.1 < 0$, thus demonstrating strong genuine multipartite entanglement.

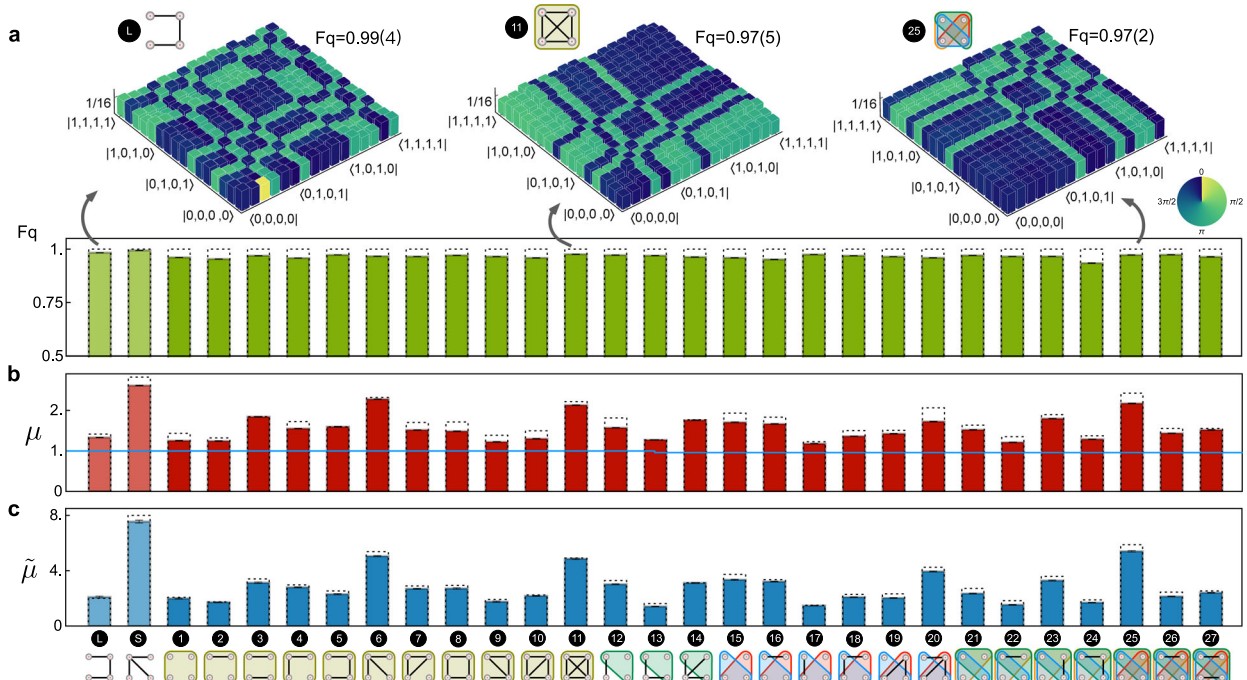

**Fig. 3 | On-chip characterization and verification of genuine multipartite entanglement in the hypergraph states. a** Measured quantum state fidelities (F_q) for all the 27 LU-equivalent classes of hypergraph states as well as 2 classes of graph states. Insets: experimentally reconstructed density matrices of three states by the quantum state topographic measurement. Column heights represent the absolute values |ρ|; colors represent the phases Arg(ρ). The values in parentheses are ±1σ uncertainty, estimated by Monte Carlo methods considering Poissonian photon statistics. **b, c** Measured Mermin parameters of μ and μ̃. Experimental results confirm the maximum violations of the Mermin–Klyshko type inequalities, indicated by the local hidden variable bound (e.g, blue line in **c**). Note the error bars for the measured F_q, μ and μ̃ are too small to be visible. Dashed lines are theoretical results.

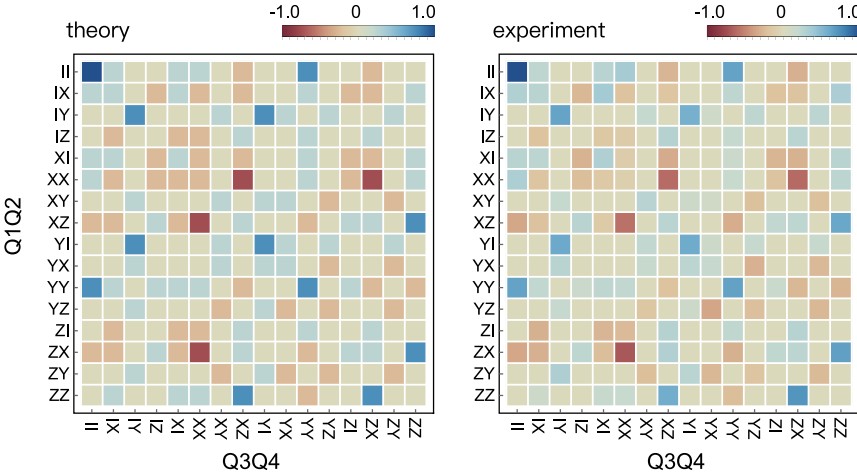

**Fig. 4 | Measurement of entanglement witness for hypergraph states.** Theoretical and experimental values for the decomposed product Pauli operators for the hypergraph state no.11. A collection of in total 256 measurements in the basis of product Pauli operators are performed. The average values for each operator are color coded and the key is provided at the top. The expected witness operator $\langle \hat{W} \rangle$ is measured to be $-0.42 \pm 0.1$.

The Mermin test provides an experimental tool to verify the presence of multiqubit entanglement that we adopt here to examine the quality of hypergraph states and the performance of the device. We measure the Mermin–Klyshko (MK) polynomials[7,36] to verify genuine multiqubit entanglement. Note that neither the detection nor distance loophole is closed in our current experiment. Mermin parameters μ and μ̃ are defined to characterize the state |ψ⟩.

$$
\begin{cases}
\mu(\psi) = \text{Max}[\langle \psi | M_n | \psi \rangle, \langle \psi | M'_n | \psi \rangle] \\
\tilde{\mu}(\psi) = \text{Max}[\langle \psi | M_n | \psi \rangle^2 + \langle \psi | M'_n | \psi \rangle^2],
\end{cases} \quad (2)
$$

where the MK polynomials $M_n$ and $M'_n$ for the $n$-qubit system are defined by $M_1 = a_1$, $M_n = \frac{1}{2} M_{n-1}(a_n + a'_n) + \frac{1}{2} M'_{n-1}(a_n - a'_n)$ (so as $M'_n$), and $\{a_i, a'_i\}$ are optimized single-qubit observables for the $i$-th qubit (see Methods). Distribution of measured Mermin parameters $\{\mu, \tilde{\mu}\}$ can provide the sufficient condition for the verification of four-qubit states as[37]: (I) If $|\psi\rangle$ is a separable state, $\mu \leq 1$. (II) If $|\psi\rangle$ is bipartite 2-entangled, i.e, having a form $|\psi_{12}\rangle \otimes |\psi_{34}\rangle$ or $|\psi_{12}\rangle \otimes |\psi_3\rangle \otimes |\psi_4\rangle$, $\tilde{\mu} \leq 2$. (III) If $|\psi\rangle$ is tripartite 3-entangled, i.e, having a form $|\psi_{123}\rangle \otimes |\psi_4\rangle$, $\tilde{\mu} \leq 4$. By violating the Mermin inequalities according to the above $\{\mu, \tilde{\mu}\}$ distribution, the entanglement structure can be verified, accordingly. For example, when the

measured $\bar{\mu}$ value is greater than 4, violating the (III) inequality, the state must be at least 4-entangled.

Figures 3b, c report the measured Mermin parameters $\mu$ and $\bar{\mu}$ for all the (27 + 2) classes of hypergraph (graph) states. When performing measurement for each state, its local projectors $\{a_i, a_i'\}$ for each qubit are optimized by an algorithm and mapped to the reconfigurable quantum chip, so as to maximize the violation of inequalities. The algorithm and a full list of optimized projections are provided in Supplementary Note 1. In Fig. 3b, the measured Mermin parameter $\mu$ for all states strongly violates the local hidden variable (LHV) classical bound. In Fig. 3c, the measured values of $\bar{\mu}$ reach more than 90% of the maximal achievable values according to quantum mechanics, confirming high-quality entanglement. The observation of $\bar{\mu}$ greater than 4, violating the (III) inequalities, e.g, for the hypergraph states labeled with a number of {6,11,20,25} and the star-type graph state labeled with S, thus identifies the genuine four-partite entanglement.

## Hypergraph states for blind MBQC

We next show the implementations of hypergraph-state MBQC for blind quantum computations[10,11,38], as illustrated in Fig. 5a. The server

capable of preparing largely entangled resource states is destined to securely share the MBQC resource states with the client who is only able to perform simple measurements to execute computation[39,40]. One main task for the client is to efficiently certificate the shared resource states, and check the correctness of quantum computing outcomes. Saying the server prepares the resource states independently in each run, a typical strategy for the client is to require some copies of the state and to perform verification before computation. These tests can return a lower bound on the average fidelity of the received states. From this result, the client can further estimate the possible error rate in the following quantum computation.

Adopting the hypergraph states as the resource states can greatly simplify the client's requirement, as its Pauli universality requires only simple Pauli measurements to be performed on qubits[10,11], making it particularly suitable for blind MBQC applications[39,40]. To benchmark the basic operations in the hypergraph-state MBQC, we choose the No.15 state in Figs. 1 and 2 (i.e, a unit cell of the Union Jack state in Fig. 5b, the resource state for Pauli universal MBQC). We perform two examples by using only Pauli measurements. In Fig. 5c, the qubit Q4 is measured in the Pauli-Y basis. It results in a measurement-induced

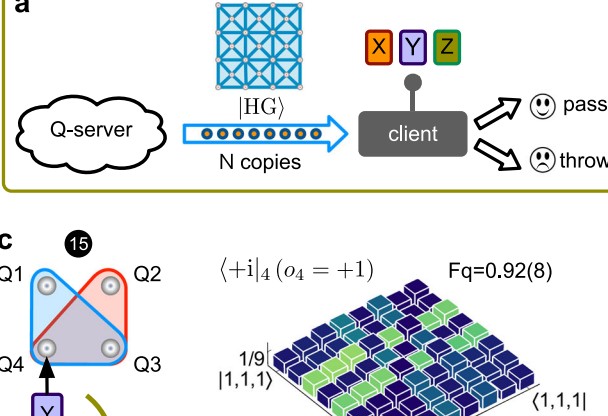

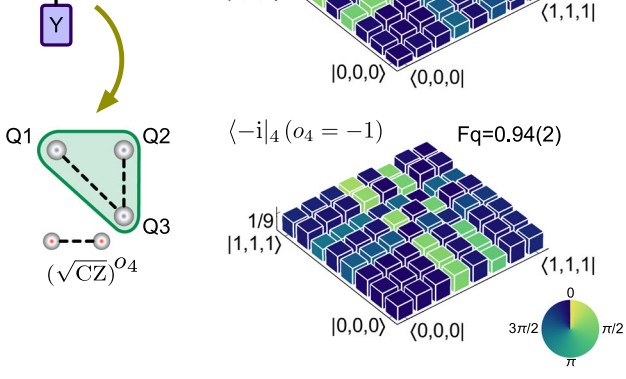

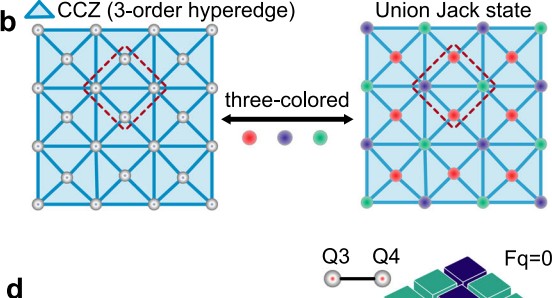

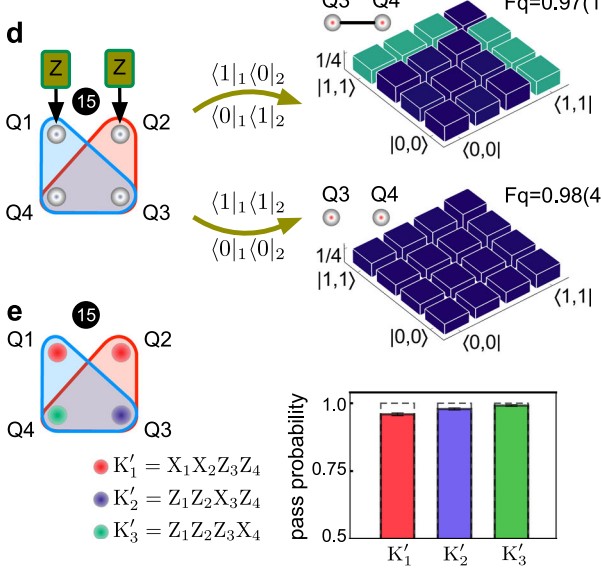

**Fig. 5 | Benchmarking the measurement-based protocol and efficient verification of hypergraph states by Pauli universal measurements. a** A diagram of verified hypergraph-state quantum computing in a blind manner between the server and the client. The server produces multipartite hypergraph states and share them with the client. For the client, only single-qubit Pauli measurements are performed for the verification of resource states sent from the server and for the execution of MBQC. **b** A two-dimensional lattice of the Union Jack state which is the resource state for Pauli universal MBQC. Three vertices of each elementary triangle are connected by an order-3 hyperedge (i.e, CCZ operation). Right plot: color-encoding stabilizers for the Union Jack state, which is 3-colored regardless of the number of qubits due to the symmetry. That means only three measurement settings are sufficient to verify the state. The group of qubits in the red dashed boxes represents a unit cell of the Union Jack state, which is the No.15 hypergraph state in Figs. 1 and 2. **c, d** Implementations of basic measurement-based protocol on the No.15 hypergraph state. In **c**, when measuring the qubit Q4 in the Y basis, an

operation of $CZ_{123}(\sqrt{CZ_{13}}\sqrt{CZ_{23}})^{o4}$ where $o4 = \pm 1$ is the measurement outcome of Q4, will add to the remaining three qubits. In **d**, it shows the state evolution when projectively measuring the qubits Q1 and Q2 in the Z basis, respectively resulting in a maximally entangled state or a product state between Q3 and Q4, determined by the outcomes of the measured qubits Q1 and Q2. In **c** and **d**, quantum operations on the remaining qubits are determined by the measurement outcomes of detected qubits. Experimentally reconstructed density matrices are shown in **c** and **d**, where column heights represent the absolute values $|\rho|$ and colors represent the phases Arg($\rho$). The $F_q$ values in parentheses are $\pm 1\sigma$ uncertainty, estimated from Monte Carlo considering Poissonian photon statistics. **e** Experimental certification of state fidelity of the Union Jack state's unit cell by the colored stabilizers. $K_i'(i = 1, 2, 3)$ represents the three colored Pauli stabilizers. A number of three measurements represented by 3-color-encoding stabilizers, regardless of the number of qubits, is sufficient to certify the state fidelity.

operation on the remaining three qubits as $CZ_{123}(\sqrt{CZ_{13}}\sqrt{CZ_{23}})^{o4}$, where $o4 = \pm 1$ is determined by the measurement outcome of Q4. Using this operation together with Hadamard gates, arbitrary quantum circuits can be simulated[12]. In Fig. 5d, measuring the qubits Q1 and Q2 in the Pauli-Z basis makes the qubits Q3 and Q4 collapse to either a maximally entangled state or a product state. When doing so on the large scale, the remaining state after measurements will be random cluster states which can be used to initialise and readout information[12].

We then show that by only performing Pauli measurements, the hypergraph states can be efficiently verified. That is, both the verification of the hypergraph resource states and their proceeding of MBQC are enabled by solely performing Pauli measurements. We adopt an approach of color-encoding stabilizers[41] to verify the No.15 hypergraph state. The idea is to perform Pauli operators that commute with the stabilizer operators of a certain group of qubits, projecting the state onto a joint eigenstate $|\psi_j\rangle$. Measurements are defined by independent sets of the hypergraph, where an independent set (denoted as $A$) is a subgroup of qubits with no connected edge between them ($\bar{A}$ is the complement set of A). Different independent sets are represented by distinct colors[41]. The operators $X_j$, $Z_k$, and $K_i$ all commute for $i, j \in A$, $k \in \bar{A}$, where $K_i = X_i \otimes \prod_{e \in E, e \ni i} C_{e/\{i\}}$ is the stabilizer for the qubit $i$, and $C_{e/\{i\}}$ corresponds to the multqubit controlled-Z gate acting on neighboring qubits around the vertex $i$ enclosed by the hyperedge $e$. When all the $X_j$ and $Z_k$ are measured, it collapses the original state into a shared eigenstate of the stabilizer $K_i$ for all $i \in A$, that is $K_i|\psi\rangle = (-1)^{t_i}|\psi_j\rangle$, where $t_i$ is given by $o_i + \sum_{e \in E, e \ni i}(\prod_{k \in e, k \neq i} o_k)$ and $(-1)^{o_i}$ is the measurement outcome of the qubit $i$. When $t_i$ equals to 1, it means that the original state is stabilized by all the $K_i$ for $i$ in set $A$, and this test on $A$ passes. The same procedure is repeated for all other independent sets. Importantly, the number of test settings depends on the number of colored independent sets to cover the entire hypergraph, instead of the number of qubits, resulting in efficient verification. For example, as shown in Fig. 5b, the Union Jack state is always a 3-color-encoding state regardless of the number of qubits. For a $s$-color-encoding hypergraph state, one colored testing set is randomly chosen in each test, and $N$ tests are repeated. If all $N$ tests pass, the verified state fidelity can be obtained as $\bar{F} \geq s \cdot \delta^{1/N} - (s-1)$, where $\delta$ is the significance level[41]. In the experiment, we measure the average value of the joint projector for each color ($K'_1, K'_2, K'_3$ in Fig. 5e). This average value equals to the passing probability $\bar{P}$ for one single test, which equivalently (see Methods) returns a lower bound of the state fidelity as $\bar{F} \geq s \cdot \bar{P} - (s-1)$. Figure 5e shows the experimental results of the Union Jack state's unit cell state, which shows high-level single-test passing probabilities with a mean value of 0.97(6). This gives a lower bound of the average state fidelity of $\bar{F} \geq 0.91$.

## Discussion

We have experimentally realized, characterized and verified complete classes of four-qubit hypergraph states on the reprogrammable photonic quantum chip. The Mermin-type inequalities and entanglement witness were characterized to verify genuine multipartite entanglement of the created hypergraph states. Basic measurement-based protocols were implemented to prototype hypergraph-state quantum computing with only Pauli measurement on single qubits. Mapping qudits to qubits enables arbitrary multiqubit-controlled gates with high fidelities and significantly improved photon rates (more than six orders higher than that in graph states[21,42,43]), thus allowing the benchmarking of hypergraph-state measurement-based quantum computation. The hypergraph-state quantum devices can be further scaled up using state-of-the-art devices and technologies, including high-efficiency photon sources[44,45], multi-photon multi-dimensional entanglement devices[24,25], large-scale silicon-photonic quantum circuits[29,46], and large-scale waveguide single-photon detectors[47].

Analysis of scalability is provided in Supplementary Note 2. Moreover, this scheme of implementing hypergraph states can be applied to other degrees of freedom of photons, such as orbital angular momentum modes[24,48,49] and frequency modes[23,50,51], which possess natural high-dimensional scalability and strong controllability. Recent technological advances in realizing high-dimensional entanglement and multi-qubit gates in superconducting[52] and trapped ion systems[53], also promise experimental implementations of the hypergraph-state quantum information processing in those systems. Our results may cast light on the investigation of complex entanglement structures in hypergraph states and the development of profound applications in Pauli-universal blind quantum computations. Though the hypergraph-state device demonstrated here is a type of noisy intermediate-scale quantum device, universal measurement-based[1,2,54] or fusion-based quantum computing[55] architectures using graph states could be upgraded to the large-scale fault-tolerant implementation of hypergraph-states quantum computing[3–6].

## Methods

### Silicon-photonic quantum chip

The chip is fabricated by standard complementary metal-oxide-semiconductor processes. The waveguide circuit patterns are defined on an 8 inches silicon-on-insulator (SOI) wafer through the 248 nm deep ultraviolet (DUV) photolithography processes and the inductively coupled plasma (ICP) etching processes. Once the waveguides layer is fabricated, a layer of silicon dioxide (SiO2) of 1$\mu$m thickness was deposited by plasma-enhanced chemical vapor deposition (PECVD). Finally, thermal-optical phase-shifters are patterned by a layer of 50-nm-thick titanium nitride (TiN) deposited on top of waveguides. Single photons were generated and guided in silicon waveguides with a cross-section of 450 nm × 220 nm. The photon-pair sources were designed with a length of 1.2 cm. Multimode interferometers (MMIs) with a width of 2.8 $\mu$m and length of 27 $\mu$m were used as balanced beamsplitters. The chip was wired-bounded on a PCB and each phase-shifter was individually controlled by an electronic driver. An optical microscopy image of the chip is shown in Fig. 2a.

### Experimental setup

In our experiment, we used a tunable continuous wave (CW) laser at the wavelength of 1550.12 nm to pump the nonlinear sources, which was amplified to 100 mW power using an erbium-doped fiber amplifier (EDFA). Photon-pairs of different frequencies were generated in integrated sources by the spontaneous four wave mixing (SFWM) process, and then spatially separated by on-chip asymmetric Mach-Zehnder interferometers (MZIs). The signal photon was chosen at the wavelength of 1545.32 nm and the idler photon at 1554.94 nm. Single-photons were routed off-chip for detection by an array of fiber-coupled superconducting nanowire single-photon detectors (SNSPDs) with an averaged efficiency of 85%, and photon coincidence counts were recorded by a multichannel time interval analyzer (TIA). The rate of photons is dependent on the choice of projective measurement bases. In the typical setting of our experiments, for example, when the state is projected to the eigenbasis, the two-photon coincidence rate was measured to be -kHz, and the integration time in the projective measurement was chosen as 5 s.

### State evolution and the $C^mZ$ gates

Our quantum photonic chip is shown in Fig. 2a, which integrates more than 400 photonic components, allowing arbitrary on-chip preparation, operation, and measurement of four-qubit hypergraph states. Key ability includes the multiqubit-controlled unitary operations $C^mU$, where $U$ represents the arbitrary unitary operation (e.g., $U = Z$ in our experiment) and $m$ is the number of control qubits. The realization of

multi-qubit $C^mU$ gates relies on the transformation from the entanglement sources to the entangling operations, by using the process of "entanglement generation – space expansion–local operation – coherent compression"[28].

Firstly, the four-dimensional Bell state is created by coherently exciting an array of four spontaneous four-wave mixing (SFWM) sources. A pair of photons with different frequencies are then separated by on-chip asymmetric Mech-Zehnder interferometers and routed to different paths, resulting in the four-dimensional Bell state[29]:

$$|\text{Bell}\rangle_4 = \frac{|0\rangle^s_{\text{qudit}}|0\rangle^i_{\text{qudit}} + |1\rangle^s_{\text{qudit}}|1\rangle^i_{\text{qudit}} + |2\rangle^s_{\text{qudit}}|2\rangle^i_{\text{qudit}} + |3\rangle^s_{\text{qudit}}|3\rangle^i_{\text{qudit}}}{2},$$
(3)

where $|k\rangle$ $(k = 0, 1, 2, 3)$ represents the logical bases of qudits, and the superscripts of $s, i$ represent the signal and idler single-photon, respectively. The two-qubit states are mapped to the four-dimensional qudit state in both of the signal and idler single-photon as the following:

$$\begin{cases} |00\rangle_{\text{qubit}} \rightarrow |0\rangle_{\text{qudit}} \\ |01\rangle_{\text{qubit}} \rightarrow |1\rangle_{\text{qudit}} \\ |10\rangle_{\text{qubit}} \rightarrow |2\rangle_{\text{qudit}} \\ |11\rangle_{\text{qubit}} \rightarrow |3\rangle_{\text{qudit}} \end{cases}$$
(4)

This results in the four-qubit state as:

$$|\Phi\rangle = \frac{|00\rangle^s_{\text{qubit}}|00\rangle^i_{\text{qubit}} + |01\rangle^s_{\text{qubit}}|01\rangle^i_{\text{qubit}}}{2} + \frac{|10\rangle^s_{\text{qubit}}|10\rangle^i_{\text{qubit}} + |11\rangle^s_{\text{qubit}}|11\rangle^i_{\text{qubit}}}{2},$$
(5)

where $|k\rangle$ $(k = 0, 1)$ represents the logical bases of qubits. For clarity, we omit the subscript of qubit in the following.

Secondly, we expand the Hilbert space of the idler-photonic qubit into a 4-dimensional space. After the space expansion process, we add two ancillary qubits $|\phi\rangle^i$ (third ququart) into the state:

$$|\Phi\rangle_1 = \frac{|00\rangle^s|00\rangle^i|\phi\rangle^i + |01\rangle^s|01\rangle^i|\phi\rangle^i + |10\rangle^s|10\rangle^i|\phi\rangle^i + |11\rangle^s|11\rangle^i|\phi\rangle^i}{2}.$$
(6)

Thirdly, the ancillary two-qubit $|\phi\rangle^i$ are locally operated using arbitrary two-qubit unitary gates represented by $U_{ij}$. We apply different unitary operations $U_{00}$, $U_{01}$, $U_{10}$, and $U_{11}$ on the $|\phi\rangle^i$ (marked by different colors in Fig. 2a). This returns a state:

$$|\Phi\rangle_2 = \frac{|00\rangle^s|00\rangle^i|\phi_R\rangle^i + |01\rangle^s|01\rangle^i|\phi_Y\rangle^i}{2} + \frac{|10\rangle^s|10\rangle^i|\phi_G\rangle^i + |11\rangle_1|11\rangle^s|\phi_B\rangle^i}{2},$$
(7)

where subscripts of {R(ed), Y(ellow), G(reen), B(lue)} represent the state after $U_{ij}$. The $U_{ij}$ are realized by universal linear-optical circuits[30].

Finally, to preserve quantum coherence, the which-process information is erased in the coherent compression process. This swaps the state information of the idler qubits as:

$$|\Phi\rangle_3 = \frac{|00\rangle^s|\phi_R\rangle^i|00\rangle^i + |01\rangle^s|\phi_Y\rangle^i|01\rangle^i}{2} + \frac{|10\rangle^s|\phi_G\rangle^i|10\rangle^i + |11\rangle^s|\phi_B\rangle^i|11\rangle^i}{2},$$
(8)

Through the post-selection procedure of projecting the last two qubits into the superposition state $(|00\rangle + |01\rangle + |10\rangle + |11\rangle)/2$, we coherently compress the 16-dimensional space back into the 4-dimensional space with a success probability of 1/4, and we obtain:

$$|\Phi\rangle_4 = \frac{|00\rangle^s|\phi_R\rangle^i + |01\rangle^s|\phi_Y\rangle^i + |10\rangle^s|\phi_G\rangle^i + |11\rangle^s|\phi_B\rangle^i}{2}.$$
(9)

In short, the process of "entanglement generation-space expansion-local operation-coherent compression" results in the multi-qubit entangling gate as:

$$|00\rangle\langle00|U_{00} + |01\rangle\langle01|U_{01} + |10\rangle\langle10|U_{10} + |11\rangle\langle11|U_{11}.$$
(10)

By reprogramming the linear-optical circuits for local unitary operations $U_{ij}$, we can realize different multi-qubits controlled unitary gates such as $C^mZ$, $m \leq 3$. For example, the triple-controlled CCCZ gate can be obtained by setting the configuration as $U_{00} = U_{01} = U_{10} = II$ and $U_{11} = CZ$. The quantum chip thus enables the generation, operation and measurement of arbitrary four-qubit hypergraph states.

## Characterizations of the CCCZ gate

We here adopt the method proposed in ref. 31 to characterize the CCCZ gate. Since the CCCZ gate is invariant with respect to the permutation of the controlled and target qubits, we can characterize the gate by measuring the input-output truth tables for four complementary product bases. In these bases, three of the qubits are prepared and measured in the computational basis states $\{|0\rangle, |1\rangle\}$ while the fourth qubit is prepared and measured in the Hadamard basis states $\{|+\rangle, |-\rangle\}$. Inputting the product state $|\psi_{i,j}\rangle$ returns a product state of $|\psi^{(\text{out})}_{i,j}\rangle = U_{CCCZ}|\psi_{i,j}\rangle$. The measured truth tables are shown in Fig. 2. We define the average statistic classical state fidelity as $F_{c(j)} = \sum_{i=1,k=1}^{16} p_{ik}q_{ik}/16$, where $p_{ik}$ and $q_{ik}$ are the theoretical and measured distribution. According to the Choi-Jamiolkowski isomorphism, we define the Choi matrix of an ideal CCCZ gate as $\chi_0$, and the experimental Choi matrix as $\chi$, from which the quantum process fidelity for the CCCZ gate can be written as $F_\chi = \text{Tr}[\chi\chi_0]/(\text{Tr}[\chi_0]\text{Tr}[\chi])$, where $\text{Tr}[\chi_0] = 16$ accounts for the normalization. We obtain the generalized Hodmann bound of fidelity[31] (the lower bounded process fidelity) for the CCCZ gate, which can be estimated from the four above averaged state fidelities as $F_\chi \geq F_{c1} + F_{c2} + F_{c3} + F_{c4} - 4$.

## Local unitary transformation

In this part, we show the rule of LU transformation when applying local Pauli operations on the hypergraph states of $|HG\rangle = (\prod_{e\in E} C_e)|+\rangle^{\otimes n}$[9], where $e$ is a hyperedge connecting vertices $\{i_1, i_2, \ldots, i_m\}$ and $C_e = I - 2(|1\rangle_{i_1}|1\rangle_{i_2}\cdots|1\rangle_{i_m}) \cdot (\langle1|_{i_1}\langle1|_{i_2}\cdots\langle1|_{i_m})$ is the corresponding multiqubit controlled-Z gates. To show the LU transformation, as an example, we consider the case when applying the Pauli $X$-operation on the $k^{th}$ qubit. The state can be written as:

$$\begin{aligned} X_k|HG\rangle &= X_k(\prod_{e\in E} C_e)|+\rangle^{\otimes n} \\ &= (\prod_{e\in E, e\not\ni k} C_e)X_k(\prod_{e\in E, e\ni k} C_e)|+\rangle^{\otimes n} \\ &= (\prod_{e\in E, e\not\ni k} C_e) \cdot [X_k(\prod_{e\in E, e\ni k} C_e)X_k]|+\rangle^{\otimes n} \\ &= (\prod_{e\in E, e\not\ni k} C_e) \cdot (\prod_{e\in E, e\ni k} X_k C_e X_k)|+\rangle^{\otimes n}. \end{aligned}$$
(11)

Now we focus on to the single operator $X_k C_e X_k$. Assume the edge $e$ connects vertices $\{1, 2, \ldots, m\}$ and for simplicity we can assume $k = 1$ is the first vertex (this does not sacrifice generality). Following the above assumption, we can write the operator explicitly as:

$$\begin{aligned} X_k C_e X_k &= X_k(I - 2|11\cdots1\rangle\langle11\cdots1|)X_k \\ &= I - 2|01\cdots1\rangle\langle01\cdots1| \end{aligned}$$
(12)

Next step we separate $C_e$ out on the left side. Notice that $I = C_e^2$ and

$$|01\cdots1\rangle\langle01\cdots1| = (I - 2|11\cdots1\rangle\langle11\cdots1|) \cdot |01\cdots1\rangle\langle01\cdots1|$$
$$= C_e|01\cdots1\rangle\langle01\cdots1| \qquad (13)$$

Therefore, we have

$$\begin{aligned}
X_k C_e X_k &= I - 2|01\cdots1\rangle\langle01\cdots1| \\
&= C_e \cdot (C_e - 2|01\cdots1\rangle\langle01\cdots1|) \\
&= C_e \cdot (I - 2|11\cdots1\rangle\langle11\cdots1| - 2|01\cdots1\rangle\langle01\cdots1|) \\
&= C_e \cdot \left(I - 2 \cdot \underbrace{(|1\rangle\langle1| + |0\rangle\langle0|)}_{I_k} \otimes \underbrace{|1\cdots1\rangle\langle11\cdots1|}_{m-1}\right) \\
&= C_e(I_k \otimes C_{e/\{k\}})
\end{aligned} \qquad (14)$$

where $C_{e/\{k\}}$ represents the multiqubit controlled gates corresponding to a new hyperedge $\{1, 2, \ldots, k-1, k+1, \ldots, m\}$.

Finally, we complete the proof by substituting the above formula into Eq.(11), which leads to

$$\begin{aligned}
X_k|G\rangle &= (\prod_{e\in E, e\not\ni k} C_e) \cdot (\prod_{e\in E, e\ni k} X_k C_e X_k)|+\rangle^{\otimes n} \\
&= (\prod_{e\in E, e\not\ni k} C_e) \cdot (\prod_{e\in E, e\ni k} C_e(I_k \otimes C_{e/\{k\}}))|+\rangle^{\otimes n} \\
&= (\prod_{e\in E} C_e) \cdot (\prod_{e\in E, e\ni k} C_{e/\{k\}})|+\rangle^{\otimes n}.
\end{aligned} \qquad (15)$$

Equation (15) shows the LU transformation rule: applying a local Pauli $X$ gate on a qubit equals to applying a series of multiqubit controlled-$Z$ gates which connect other qubits that share the same edge with it.

We take an example to illustrate local unitary transformation, as shown in Fig. 1c. The initial state is

$$\begin{aligned}
|\psi\rangle = &|0000\rangle + |0001\rangle + |0010\rangle + |0011\rangle \\
&+ |0100\rangle - |0101\rangle + |0110\rangle + |0111\rangle \\
&+ |1000\rangle + |1001\rangle + |1010\rangle + |1011\rangle \\
&- |1100\rangle - |1101\rangle - |1110\rangle - |1111\rangle
\end{aligned} \qquad (16)$$

After applying $X3$, which flips the third qubit, the state becomes

$$\begin{aligned}
|\psi\rangle = &|0000\rangle + |0001\rangle + |0010\rangle + |0011\rangle \\
&+ |0100\rangle + |0101\rangle + |0110\rangle - |0111\rangle \\
&+ |1000\rangle + |1001\rangle + |1010\rangle + |1011\rangle \\
&- |1100\rangle - |1101\rangle - |1110\rangle - |1111\rangle
\end{aligned} \qquad (17)$$

which can be quickly verified as the expression for the second hypergraph state in Fig. 1c. Following a similar procedure, the hypergraph can be simplified to only two edges as shown in Fig. 1c. The rule of LU transformation can be graphically described as the $X(k)$ operation on the qubit $k$ removes or adds these hyper-edges in $E^{(k)}$ depending on whether they exist already or not, where $E^{(k)}$ represents all hyper-edges that contain qubit $k$ but removing qubit $k$ out. The $Z(k)$ operation on the qubit $k$ remove the one-edge on the qubit $k$.

## Measurement basis for the Mermin–Klyshko (MK) polynomials
We here derive the basis used for the evaluation of MK polynomials $M_4$ and $M_4'$. The general form of $M_n$ is given as[37]:

$$M_n = \frac{1}{2}M_{n-1}(a_n + a_n') + \frac{1}{2}M_{n-1}'(a_n - a_n') \qquad (18)$$

where $a_n$ and $a_n'$ are single-qubit operators and $M_1 = a_1$. $M_n'$ can be obtained by interchanging the terms with and without the prime. In particular, for the four-qubit state, we then have $M_4$ and $M_4'$:

$$\begin{cases} M_4 = \frac{1}{2}M_3(a_4 + a_4') + \frac{1}{2}M_3'(a_4 - a_4') \\ M_4' = \frac{1}{2}M_3(a_4 + a_4') - \frac{1}{2}M_3'(a_4 - a_4'). \end{cases} \qquad (19)$$

Similarly, $\{M_3, M_2\}$ and $\{M_3', M_2'\}$ can be obtained. We instead use an alternative way by dividing the original 4-qubit operators into 2-qubit by 2-qubit parts because of the implementation of qubit-qudit mapping in our device. This leads to the construction of the MK polynomials $M_4$ and $M_4'$ from $M_2$ and $M_2'$:

$$\begin{cases} M_4 = \frac{1}{2}[M_2(a_3 a_4' + a_3' a_4) + M_2'(a_3 a_4 - a_3' a_4')] \\ M_4' = \frac{1}{2}[M_2'(a_3 a_4' + a_3' a_4) - M_2(a_3 a_4 - a_3' a_4')]. \end{cases} \qquad (20)$$

In experiment, we first measured the $M_2, M_2', (a_3 a_4' + a_3' a_4)$ and $(a_3 a_4 - a_3' a_4')$, and then estimated the MK polynomials $M_4$ and $M_4'$. A total number of 64 bases are required for $M_4$ and $M_4'$, each of which is determined by the choice of the corresponding $a_i$ and $a_i'$.

## Efficient verification of hypergraph states
In blind quantum computation, clients use the expensive resource states shared by the server to perform their measurements. In such a scenario, the average fidelity of the states generated by the server has to be verified before computation. Ideally, the clients are capable of estimating a lower bound of the state fidelity and verifying genuine entanglement, without much cost. We here use a protocol of color-encoding stabilizers[41]. To achieve a verification of fidelity larger than $1 - \epsilon_0$, the number of states required is given by

$$N = \left\lceil \frac{\ln(\delta)}{\ln(1 - \epsilon_0/s)} \right\rceil, \qquad (21)$$

where $s$ is the minimum number of colors in the hypergraph state, $\delta$ is the significance level and $\epsilon_0$ denotes the error. This formula can be better understood in the following form

$$\delta \geq (1 - \epsilon_0/s)^N, \qquad (22)$$

where the right-hand side represents a total passing probability of the total $N$ tests for a state with an infidelity $\epsilon_0$. When this probability is smaller than the chosen significance level and a passing event occurs on the client side, we can draw the conclusion that the real infidelity of the state generated from the server should satisfy $\epsilon < \epsilon_0$ with a significance level $\delta$.

A simple transformation of Eq. (21) gives

$$\bar{F} \geq s \cdot \delta^{1/N} - (s - 1). \qquad (23)$$

In the ideal case, if the generated state is exactly the target hypergraph state, i.e, $F = 1$, the probability of passing the test is always 100%, while increasing the number of tests will result in a tighter bound (smaller $\epsilon_0$). In reality, for experimental states with non-unit fidelity, the total passing probability will decrease exponentially with the number of tests $N$. When we define the single-test passing probability as $\bar{P}$, the total passing probability will take the form of $\bar{P}^N$, which should be kept above the significance level $\delta$. Therefore, for a selected significance level, the maximum number of tests, which corresponds to the tightest bound on fidelity, should satisfy $\bar{P}^N = \delta$. Replacing $\delta$ by $\bar{P}^N$ in Eq. (23) thus returns

$$\bar{F} \geq s \cdot \bar{P} - (s - 1). \qquad (24)$$

## Data availability
The data that support the findings of this study are available from the corresponding author upon request.

## Code availability

The code that support the findings of this study are available from the corresponding author upon request.

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

## Acknowledgements

We acknowledge support from the Natural Science Foundation of China (nos 12325410, 62235001, 61975001), the Innovation Program for Quantum Science and Technology (no. 2021ZD0301500), the National Key R&D Program of China (no. 2019YFA0308702), and Beijing Natural Science Foundation (Z190005, Z220008).

## Author contributions

J.W. conceived the project. J.H. and X.L. contributed equally to this work. J.H., X.L., and Y.C. built the setup and carried out the experiment. J.H., X.L., X.C., C.Z., and Y.Z. performed the theoretical analysis. Y.L., Q.H., Q.G., and J.W. managed the project. All authors discussed the results and contributed to the manuscript.

## Competing interests

The authors declare no competing interests.
