## [Peer Review File · Nature Communications]

Demonstration of hypergraph-state quantum information processingEditorial Note: This manuscript has been previously reviewed at another journal that is not operating a transparent peer review scheme. This document only contains reviewer comments and rebuttal letters for versions considered at *Nature Communications*.

REVIEWERS' COMMENTS

Reviewer #1 (Remarks to the Author):

In light of the changes made according to my previous comments, I find that the manuscript has substantially improved both in its clarity and readability. I am happy to recommend the article for publication in Nature Communications.

Reviewer #2 (Remarks to the Author):

The manuscript explores the experimental creation of four-qubit quantum hypergraph states, representing a significant step forward in quantum information processing. The authors delve into various considerations, including gate fidelity, nonlocality, blind state verification, use in BMQC, and certification using colored stabilizers, implementing a range of results from existing theories.

The paper is well-written, and the experimental work on creating hypergraph states using a quantum photonic chip is commendable. The concise presentation of the implementation of CZ, CCZ, or CCCZ gates with high fidelity in the Methods section is a notable feature, demonstrating the authors' expertise in this field.

While the manuscript primarily builds on existing literature, the practical application to the simplest nontrivial example adds value to the field. The authors successfully leverage a quantum photonic chip for on-chip preparation, operation, and measurement of four-qubit hypergraph states.

A previous suggestion for providing a more in-depth discussion of experimental losses and potential scalability issues in the system was taken into account. Addressing these aspects enhanced the manuscript's completeness and practical applicability. Moreover, the authors clearly indicated that they are not closing any loopholes in Bell-type experiments.

Bell's experiments with strong loopholes (as in this manuscript) put the entire consideration

of the setup in question, but the goal of the paper appears to showcase the strength of the generated entanglement and not to disprove local realistic theories as it happens often in the Bell tests. Still, it is not clear how robust such nonlocal properties will remain in real or near-term applications. However, this can be considered in the follow-up reports.

To summarise, I am inclined to recommend the manuscript for publication in Nature Communications.

Reviewer #3 (Remarks to the Author):

The authors have performed substantial revisions on their manuscript according to my previous reports, and to the reports of the other Reviewers. More specifically, the authors have amended the manuscript in several directions following my advices. These modifications include adding a significant amount of additional relevant information on the integrated device, and on the generation and measurement protocol. This enables now a clear understanding of the system operation and performances. Additionally, and importantly, the authors have also clarified other two aspects regarding the novelty/motivation of their work, and the scalability towards future larger implementations. In the new version of the manuscript, this has been significantly improved and enables a better understanding of the relevance of the reported results.

Given the new version of the manuscript, I can provide a positive recommendation of this paper for publication in Nature Communications.